# The Effect of the Classical TSPO Ligand PK 11195 on In Vitro Cobalt Chloride Model of Hypoxia-like Condition in Lung and Brain Cell Lines

**DOI:** 10.3390/biom12101397

**Published:** 2022-09-29

**Authors:** Rula Amara, Nidal Zeineh, Sheelu Monga, Abraham Weizman, Moshe Gavish

**Affiliations:** 1Ruth and Bruce Rappaport Faculty of Medicine, Technion-Israel Institute of Technology, Haifa 31096, Israel; 2Sackler Faculty of Medicine, Tel Aviv University, Tel Aviv 6997801, Israel; 3Research Unit, Geha Mental Health Center and Felsenstein Medical Research Center, Petah Tikva 4910002, Israel

**Keywords:** hypoxia-like condition, H1299, BV-2, CoCl_2_, PK 11195

## Abstract

The mitochondrial translocator protein (TSPO) is a modulator of the apoptotic pathway involving reactive oxygen species (ROS) generation, mitochondrial membrane potential (Δψm) collapse, activation of caspases, and eventually initiation of the apoptotic process. In this in vitro study, H1299 lung cells and BV-2 microglial cells were exposed to the hypoxia-like effect of CoCl_2_ with or without PK 11195. Exposing the H1299 cells to 0.5 mM CoCl_2_ for 24 h resulted in decreases in cell viability (63%, *p* < 0.05), elevation of cardiolipin peroxidation levels (38%, *p* < 0.05), mitochondrial membrane potential depolarization (13%, *p* < 0.001), and apoptotic cell death (117%, *p* < 0.05). Pretreatment with PK 11195 (25 µM) exhibited significant protective capacity on CoCl_2_-induced alterations in the mentioned processes. Exposure of BV-2 cells to increasing concentrations of CoCl_2_ (0.3, 0.5, 0.7 mM) for 4 h resulted in alterations in the same cellular processes. These alterations were obtained in a dose-dependent manner, except the changes in caspases 3 and 9. The novel ligands as well as PK 1195 attenuated the in vitro hypoxia-like effects of CoCl_2_. It appears that the TSPO ligand PK 11195 can prevent CoCl_2_-induced cellular damage in both non-neuronal and brain cell lines, and they may offer a novel approach to the treatment of hypoxia-related lung and brain diseases in some cases that fail to respond to conventional therapies.

## 1. Introduction

Hypoxia refers to an abnormal exposure of cells or tissues to low level of oxygen. The cellular response to hypoxia has been of great interest to researchers since hypoxia is relevant to essential biological processes including angiogenesis, cellular survival/proliferation, energy metabolism, erythropoiesis, extra-cellular matrix function, invasion/metastasis, iron metabolism, pH regulation, multi-drug resistance, and stem cell survival [1,2].

Neurodegenerative diseases are defined by the progressive loss of neurons, synapses, and protein misfolding, and aggregation of proteins [3]. Reduced oxygen supply has been suggested as an important contributor to pathogenesis of neurodegenerative diseases. Hypoxia was found to induce oxidative stress, inflammation, and apoptosis, among other cellular processes, contributing to the pathophysiology of neurodegeneration [3]. 

CoCl_2_ is a hypoxia-mimicking agent [4]. It inhibits prolyl hydroxylase-domain enzymes (the oxygen sensors) through replacement of Fe^2+^ with Co^2+^ making these enzymes unable to label hypoxia-inducible factor (HIF)-alpha for degradation [5,6]. It should be noted that CoCl_2_ mimics HIF-1 alpha accumulation, but no other effects of hypoxia. In addition to its effects on HIF-1 alpha accumulation, exposure to CoCl_2_ can also modulate processes and pathways, such as apoptosis and reactive oxygen species (ROS) generation [7,8]. Notably, exposure to CoCl_2_ is not associated with a decrease in pO_2_, thus, it actually represents an hypoxia-like condition [9].

The mitochondrial translocator protein (TSPO) was shown to regulate ROS generation, proliferation, angiogenesis, and apoptosis, cellular processes that are relevant to the toxic effect of CoCl_2_ [9,10].

TSPO is highly expressed during inflammation and in various tumor types, such as glioblastomas [11]. In the U118MG glioblastoma cell line, exposure to CoCl_2_ at various concentrations was shown to increase ROS formation, cause the collapse of the mitochondrial membrane potential and cause cell death [12,13,14,15]. TSPO knockdown using siRNA, or its blockade using the TSPO antagonist PK11195, significantly counteracted the CoCl_2_-induced effects [9]. Several studies investigated CoCl_2_-induced hypoxia using different cellular models [16,17,18]. Our research group evaluated the effects of TSPO ligands on CoCl_2_-induced cytotoxicity, on the human H1299 lung cancer, and glial cell line. These cell lines were used as cellular models of response of brain and lung tissues in hypoxic states. Further investigations on this topic may produce implications on the role of TSPO as a candidate target for the treatment of hypoxic conditions. 

In the present study, we evaluated the protective effects of the TSPO ligand PK 11195 in non-neuronal (lung cancer cell line) and brain (microglial cell line) cells exposed to the hypoxic agent CoCl_2_ [17] in an attempt to identify the cellular mechanisms that are involved in such putative beneficial effects in peripheral and brain cells. In all in vitro experiments, we used PK 11195 at the same concentration of 25 µM in both cell lines, since the affinity of [^3^H] PK 11195 to TSPO is at the nanomolar range. Furthermore, the density of TSPO is between 3000–4000 fmol/mg protein in both H1299 lung cancer cells and BV-2 microglial cells [19,20]. 

## 2. Methods

### 2.1. Cell Lines

In this study, we used two cell lines: (1) Human H1299 lung cancer cell line from the American Type Culture Collection (ATCC). The culture medium used consisted of RPMI (high glucose, with no L-Glutamine and no Sodium Pyruvate), supplemented with Fetal Bovine Serum (10%), 2 mM L-Glutamine and gentamycin (50 µg/mL). (2) Murine BV-2 microglial cell line (generously provided by Zvi Vogel’s laboratory, Weizmann Institute of Science, Rehovot, Israel): The BV-2 cells were cultured in Dulbecco’s modified Eagle’s medium high glucose containing 4.5 g/L glucose, 4 mM L-glutamine and supplemented with 5% fetal bovine serum, penicillin (100 U/mL), and streptomycin (100 µg/mL) 2018. These two cell types were cultured at 37 °C in 5% CO_2_ and 90% relative humidity.

### 2.2. Tspo Ligands Pretreatment

The in vitro experiments included the following groups: vehicle control group pretreated with 1% ethanol-containing 0.5% Fetal Calf Serum (FCS, biological industries, Beit Ha’Emek, Israel); a group pretreated with 1% ethanol-containing serum starvation medium. 

The two cell lines were seeded in 96-well plates (5 × 10^3^ cells per well) or 6-well plates (2.5 × 10^5^ cells per well) (depending upon the type of experiment) and grown in complete medium for 48 h at 37 °C and 5% CO_2_ until the desired confluency 80% was reached. Then, pretreatment with the TSPO ligand PK 11195 (25 µM) in serum-starvation medium was applied for another 24 h. In all experiments, we used PK 11195 at a concentration of 25 µM since it was found to affect cellular functions relevant to hypoxia [9].

### 2.3. Cobalt Chloride (CoCl_2_) Exposure

CoCl_2_ (Sigma-Aldrich, Rehovot, Israel) was prepared at the concentrations required for each specific experiment and applied to the CoCl_2_-treated groups for 24 h [16,17].

### 2.4. Cell Counting 

Cells were grown until 80% confluency was reached, the medium was discarded, cells were washed with phosphate buffer saline (PBS) and collected following trypsinization. For cell counting, 100 µL of the cells was placed in an Eppendorf tube then mixed with 100 µL of trypan blue (ratio 1:1). Under the light microscope, the cells were counted by hemocytometer (Neubauer slide, Sigma Aldrich, Rehovot, Israel).

### 2.5. XTT Assay 

The two cell lines (H1299 and BV-2) were seeded in 96-well plates (5000 cells/well) and grown for 48 h in complete medium. Then, they were pretreated with the required TSPO ligand for another 24 h, followed by exposure of the cells to the desired concentrations of CoCl_2_ for 30 min, 4 h, or 24 h. Assessment of cellular viability was performed using a XTT cell viability kit (Biological Industries, Bait Ha’Emek, Isreal), following the manufacturer’s protocol:150 µL medium from each well was removed followed by adding 50 µL from the XTT mixture to the remaining 50 µL medium within the plate, then the plates were incubated in dark for one hour and a half. Reduction of 2,3-bis-(2-methoxy-4-nitro-5-sulfophenyl)-2H-tetrazolium-5-carboxanilide (XTT) depends on mitochondrial dehydrogenases and reductases, which results in orange formazan dye production, a process that occurs only in viable cells. The amount of the orange dye indicates the cellular viability and the optic density (O.D.) was measured using Infinite M200 Pro plate reader (Tecan, Männedorf, Switzerland) with absorbance with an endpoint photometric at 492 nm wavelength and reference wavelength of 620 nm.

### 2.6. JC-1 Assay 

The mitochondrial membrane potential (Δψm) depolarization was assessed using the JC-1 assay, which was based on cationic, lipophilic tetra-ethyl-benzimidazolyl-carbocyanine iodide JC-1 dye. After seeding the H1299 cells in 6-well plates (250,000 cells/well) for 48 h and pretreated with 25 µM PK 11195 for another 24 h, cells were exposed to 0.5 mM CoCl_2_ for 24 h, then cells were trypsinized (600 µL trypsin), collected, and centrifuged (660× *g* for 5 min at room temperature) followed by removing the supernatant and resuspended in 600 µL of PBS and cells were centrifuged again. Dilution of JC-1 with PBS (1:500) was carried out and 600 µL of the solution was applied and incubated for 30 min in the dark. Cells were centrifuged again and 400 µL of PBS were added followed by cells filtration and transfer to FACS tubes. In case of intact cells with high Δψm, JC-1 enters the mitochondria and forms J- aggregates emitting red fluorescence at 590 nm. In contrast, cells exposed to CoCl_2_, with low Δψm, the JC-1 dye remain in the cytosol compartment in a monomer form emitting green fluorescence at 527 nm indicating Δψm depolarization [9]. The median fluorescence intensity (MFI) indicates Δψm depolarization that was calculated by red/green ratio using FACS, and the results were analyzed using FlowJo (FlowJo LLC, Ashland, Oregon) 

### 2.7. Nonyl Acridine Orange (NAO) Assay-Cardiolipin Peroxidation Indicator

H1299 Cells were seeded in 6-well plates (250,000 cells/well), after 48 h, 25 µM of PK 11195 was applied for 24 h, then cells were exposed to 0.5 mM CoCl_2_ for another 24 h. Afterwards, cells were trypsinized and centrifuged (660× *g* for 5 min at room temperature) followed by neutralizing trypsin with complete medium. The cells were washed with PBS and centrifuged again, the supernatant was aspirated, and 400 µL of Nonyl Acridine Orange (NAO) stain (diluted with PBS at a ratio of 1:1000) was added. The cells were incubated in the dark for 30 min, and centrifuged again, then 400 µL of PBS was added and transferred to FACS tubes. 

NAO stain was used to assess the cardiolipin peroxidation level. Cardiolipin, a polyunsaturated acidic phospholipid, biosynthesized and localized in the inner mitochondrial membrane. Cardiolipin is known to have high content of unsaturated fatty acid, which makes it more susceptible to ROS-related cardiolipin peroxidation that results in cytochrome c translocation to the cytosolic compartment, where it initiates the mitochondrial apoptotic cascade. NAO staining of the cardiolipin content was performed as described previously. Elevated MFI of NAO staining indicates lower cardiolipin content due to increased cardiolipin peroxidation. The MFI was measured using CyAN ADP FACS machine (Beckman Coulter, Brea, CA, USA), and the results were analyzed using FlowJo (10th version, FlowJo LLC, Ashland, OR, USA). 

### 2.8. Necrosis/Apoptosis Assay 

A necrosis/apoptosis assay kit (Abcam, Cambridge, UK) was used to detect apoptosis and necrosis levels in H1299 cells according to the manufacturer’s instructions. Seeding of H1299 cells was performed in 6-well plates (250,000 cells/well). After 48 h, PK 11195 at a concentration of 25 µM, was applied for 24 h then followed by exposure of 24 h to 0.5 mM of CoCl_2_. Cells were trypsinized using 600 µL trypsin (Trypsin EDTA Solution B (0.25%), EDTA (0.05%)) (Biological industries, Beit Ha’Emek, Israel) centrifuged (660× *g* for 5 min at room temperature) then the supernatant was discarded, and the cells were resuspended in 200 µL assay buffer (provided in the kit) and transferred to Eppendorf tubes. Staining was conducted by adding 2 µL of Apopoxin to 100 µL of sample for apoptotic cells detection, 1 µL of 7-AAD to 100 µL of sample for necrotic cells detection with subsequent incubation in the dark for 1 h. Before reading the samples, 300 µL of assay buffer were added to each sample. An Aria FACS machine (BD bioscience, San Jose, CA, USA) was used with Ex/Em = 490/525 nm for detection of apoptosis, Ex/Em = 550/650 nm for detection of necrosis. The 10th version of FlowJo (LLC, Ashland, OR, USA) was used for calculation. 

### 2.9. Cellular ROS/Superoxide Assay

A cellular ROS/superoxide assay kit (Abcam, Cambridge, UK) was used to detect ROS levels in BV-2 cells upon exposure to 0.3, 0.5, and 0.7 mM CoCl_2_ for 4 h and 24 h, according to the manufacturer’s instructions. On the day of the experiment, the medium was removed from wells and washed with 1X washing buffer (provided in the kit in 10× concentration). A total of 100 µL/well of ROS detection dye was applied on the treatment groups for 1 h in the dark at 37 °C. Bottom reading of the plates using Infinite M200 Pro plate reader (Tecan, Männedorf, Switzerland) was performed with Ex/Em = 488/520 nm. 

### 2.10. Caspases Multiplex Activity Assay 

Assessment of the apoptotic markers caspase 3 and caspase 9 was performed in BV-2 cells using the caspase multiplex activity assay kit (Abcam, Cambridge, UK). BV-2 cells were seeded in 96-well plates (10^4^ cells/90 µL) and pretreated as required, following the cells exposure to the desired concentrations of CoCl_2_ (0.5 and 0.7 mM) for 4 h. The diluted caspases 3 and 9 substrates (diluted in assay buffer at ratio of 1:200) were added to the cells and incubated for one hour. Measurements of fluorescence intensity at Ex/Em of 535/620 nm for caspase 3 and at Ex/Em of 370/450 for caspase 9 were conducted using the Infinite M200 Pro plate reader (Tecan, Männedorf, Switzerland). 

### 2.11. TSPO Protein Expression Levels 

Measurement of TSPO protein levels in BV-2 cells was performed using FACS upon 4 h of exposure to 0.3, 0.5, and 0.7 mM CoCl_2_. Medium with the cells were collected from each sample and centrifuged (660× *g* for 5 min at room temperature), then supernatant was removed. Cells were washed using PBS and incubated with PBS containing 0.2% Tween 20 (PBS-T) for 10 min and centrifuged again at 660× *g* for 5 min at room temperature. Then, the cells were incubated overnight with primary anti-TSPO antibody (Abcam, Cambridge, UK) diluted with ratio 1:100 in 3% BSA in PBS-T. On the following morning, the cells were washed with PBS, incubated with goat anti-rabbit AlexaFlour488 secondary antibody for 1 h, then washed and transferred to FACS tubes. MFI was measured using FACS machine. The results were analyzed using FlowJo 10th version (LLC, Ashland, OR, USA).

### 2.12. Statistical Analyses

Results are presented as mean ± standard deviation (SD). One-way analysis of variance (ANOVA) test was used for comparisons among groups, followed by Bonferroni’s post-hoc test. Statistical significance was defined by *p* < 0.05. At least 4 or more biological replicates were used for each group in all experiments.

## 3. Results

The protective ability of PK 11195 to counteract the effect of CoCl_2_-induced mitochondrial damage via TSPO-related processes in non-neuronal (lung cancer) and central (microglial) cell lines included the assessment of: cell viability, oxidative stress, mitochondrial membrane potential depolarization, TSPO protein expression levels, and apoptotic markers. 

### 3.1. H1299 Lung Cancer Cell Line

#### 3.1.1. The Impact of CoCl_2_ on Cellular Viability Using XTT Assay

Dose–response analysis of cellular viability was measured in H1299 cells after exposure for 24 h to varying concentrations of CoCl_2_ (0.1–1 mM). Significant decreases in cell viability were detected at 0.3 to 1 mM of CoCl_2_. Since a sufficient, but not complete suppressive effect of CoCl_2_ on cellular viability was observed at 0.5 mM, this concentration was chosen for all the further experiments (*p* < 0.001; Figure 1).

#### 3.1.2. The Impact of TSPO Ligands on CoCl_2_-Induced Cellular Death (Using XTT Assay)

Analysis of the protective capacity of the TSPO ligand PK 11195 at a concentration of 25 µM upon H1299 cells exposed to 0.5 mM of CoCl_2_ is shown in Figure 2. H1299 cells were exposed to CoCl_2_ (0.5 mM) with or without PK 11195. Following exposure to CoCl_2_, 80.3% cells remained viable in CoCl_2_+PK 11195 (*p* < 0.001; Figure 2). In contrast, the group treated with CoCl_2_ 46.3% cells remained viable (*p* < 0.01; Figure 2).

#### 3.1.3. The Impact of PK 11195 on CoCl_2_-Induced Oxidative Stress (Using NAO Assay)

Cardiolipin peroxidation levels were determined by FACS. Exposure of H1299 cells to 0.5 mM CoCl_2_ induced a decrease in NAO fluorescent intensity by 38% (*p* < 0.05 vs. control). Treatment with 25 µM PK 11195 24 h prior to exposure to 0.5 mM CoCl_2_, almost completely prevented the CoCl_2_-induced decrease in NAO fluorescence intensity (Figure 3A,B).

#### 3.1.4. The Impact of PK 11195 on CoCl_2_-Induced Mitochondrial Membrane Potential Depolarization (Using JC-1 Assay)

Exposure of H1299 cells to 0.5 mM CoCl_2_ resulted in a significant mitochondrial membrane potential depolarization decrease by 13% (*p* < 0.001) as compared to the control group. Pretreating the cells with PK 11195 at a concentration of 25 µM completely inhibited the CoCl2-induced membrane potential depolarization (Figure 4A,B). Figure 4C expresses the unstained cells group, demonstrating cells unexposed to CoCl_2_ and not stained with JC-1 dye.

#### 3.1.5. The Impact of PK 11195 on CoCl_2_-Induced Cellular Damage: Apoptotic and Cell Death

Exposing H1299 cells to 0.5 mM CoCl_2_ for 24 h led to a significant rise in apoptotic cell death by 117% (*p* < 0.05) as compared to the control group. Pretreatment with 25 µM of PK 11195 showed the capacity (*p* < 0.05) to counteract the CoCl_2_-induced apoptosis (Figure 5). PK 11195 alone did not show any significant difference compared to the control.

### 3.2. BV-2 Glial Cell Line

#### 3.2.1. The Toxic Effect of Cocl_2_: Cellular Viability Using Xtt Assay

A dose–response study was carried out of cellular viability for BV-2 microglial cells exposed to CoCl_2_ (0.1,0.3, 0.5, 0.7, 0.9 mM) for 30 min, 4 h and 24 h. After 30 min of exposure to the various concentrations of CoCl_2_, significant decreases in cell viability were observed at 0.7 and 0.9 mM by 20% (*p* < 0.05) and 29% (*p* < 0.001), respectively. After 4 h of exposure to CoCl_2_, significant reductions were observed at 0.5, 0.7, and 0.9 mM by 14% (*p* < 0.01), 32% (*p* < 0.001), and 44% (*p* < 0.001), respectively. After 24 h of CoCl_2_ exposure, a significant concentration-dependent decrease in cell viability by 19% (*p* < 0.001) at 0.1 mM and reached maximal reduction at 0.9 mM by 90% (*p* < 0.001; Figure 6).

#### 3.2.2. The Impact of PK 11195 on CoCl_2_-Induced Oxidative Stress Using Cellular ROS/Superoxide Detection Assay Kit

Measurements of oxidative stress levels in BV-2 cells following exposure for 4 h and 24 h to CoCl_2_ at 3 different concentrations (0.3, 0.5 and 0.7 mM) with and without the TSPO ligand PK11195 at a concentration of 25 µM. Significant increase by 26% (*p* < 0.001) in oxidative stress levels was detected following 4 hours’ exposure of BV-2 cells to 0.7 mM CoCl_2_ as compared to control group_._ PK11195 (25 µM) did not prevent this increase in oxidative stress (Figure 7A). Exposure of cells to CoCl_2_ (0.3, 0.5, and 0.7 mM) for 24 h resulted in significant increases in oxidative stress levels by 117%, 198% and 210% (*p* < 0.001 for all), respectively. Pretreatment with PK 11195 (25 µM) significantly attenuated this increase in oxidative stress induced by CoCl_2_ exposure for 24 h (*p* < 0.001; Figure 7B).

#### 3.2.3. The Impact of PK 11195 on CoCl_2_-Induced Changes in TSPO Protein Expression Using FACS

A significant elevation by 42% (*p* < 0.05) of TSPO protein expression levels was obtained following exposure to 0.7 mM CoCl_2_, as compared to the control group. Pretreatment with 25 µM PK 11195, totally prevented the CoCl_2_-induced TSPO expression elevation following exposure to 0.7 mM CoCl_2_ (*p* < 0.001; Figure 8).

#### 3.2.4. The Impact of PK 11195 Pretreatment on CoCl_2_-Induced Cellular Apoptosis: Caspases 3 and 9 Using Caspases Multiplex Activity Assay Kit

Evaluation of caspases 3 and 9 levels after exposing BV-2 cells for 4 h to various concentrations of CoCl_2_ with or without 25 µM PK 11195 pretreatment. Only at 0.7 mM, CoCl_2_ induced significant increase in caspase 3 levels by 196% (*p* < 0.001), PK 11195 prevented this elevation by 83% (*p* < 0.01; Figure 9A). The levels of caspase 9 were not affected by CoCl_2_ or by the presence of PK 11195.

## 4. Discussion

In the present study, the capacity of the TSPO ligand PK 11195 to counteract the CoCl_2_-induced hypoxia-like cellular damage was evaluated in both non-neuronal (lung) and central (microglia) cell lines.

### 4.1. Lung Cancer Cell Line (H1299)

We used an established in vitro model of pulmonary hypoxia-like condition by exposing the H1299 lung cancer cell line to cobalt chloride and investigated the protective effects of TSPO ligands in this cellular model [16,17]. According to our data, lung derived cells exposed to various concentrations of CoCl_2_, ranging from 0.1 mM to 1 mM, for 24 h led to a dose-dependent reduction in cell viability with maximal toxicity at 1 mM as compared to a control group. Based on the current dose-dependent results, and similar to previous studies, [10] which showed sufficiently effective toxic damage that mimics hypoxia-like condition at a concentration of 0.5 mM of CoCl_2_, in our experiments we assessed the impact of this concentration on a variety of TSPO-related cellular processes, including: cell death, mitochondrial membrane potential depolarization, cardiolipin peroxidation, and ROS generation. We evaluated the protective effects of the TSPO ligands PK 11195 at a concentration of 25 µM in the CoCl_2_ hypoxia-like cellular model. PK 11195 exhibited a significant inhibitory effect on CoCl_2_-induced cell viability reduction.

TSPO has been reported to be involved in several mitochondrial processes affected by CoCl_2_ exposure, including apoptosis, ROS generation, and collapse of mitochondrial membrane potential. One putative pathway for the CoCl_2_-induced cytotoxicity is through a harmful impact on mitochondrial functions mediated by TSPO, including CoCl_2_-induced apoptosis mediated by ROS generation, cardiolipin peroxidation, mitochondrial membrane potential depolarization, and decreased cellular metabolism and viability.

In our present study, using ROS generation as an indicator of oxidation, PK 11195 exhibited a potent inhibitory effect on CoCl_2_-induced cardiolipin peroxidation. These findings further strengthen the previously published data on U118MG cells as a model for glioma [16].

It is likely that the CoCl_2_-induced accumulation of ROS interfered with the mitochondrial homeostasis of the mitochondrial membrane potential. Indeed, a significant increase was seen in depolarization of the mitochondrial membrane potential in CoCl_2_-treated cells as compared to the unexposed control group. It appears that CoCl_2_ led to ROS generation and disruption of mitochondrial potential which eventually led to cytochrome c release and subsequent initiation of apoptotic pathway [19,21]. A previous microscopic study revealed morphological, nuclear, and cytological changes as features of apoptosis including condensed chromatin, DNA fragmentation, cell shrinkage, and cell surface blabbing [10,20]. In this study, necrosis/apoptosis assay was used to assess the harmful effects of CoCl_2_ on cell viability (apoptotic or necrotic cell death). Following exposure to CoCl_2,_ H1299 cells showed significant elevation in apoptosis levels, but not necrosis, as was reported previously [16,19,21]. Notably, another study performed by our group using cigarette smoke as a hypoxia-causing agent leading damage to cellular hypoxia, also resulted in apoptotic cell death, rather than necrotic cell death [19,21].

In the current study, PK 11195 exhibited a significant inhibitory effect on CoCl_2_-induced cell viability reduction. A previous study demonstrated the efficacy of the classical TSPO ligand PK 11195 in counteracting the effect of the CoCl_2_-induced damages in astrocytic cell line (U118MG) [16]. In another study, in the same H1299 lung derived cells, the classical high affinity TSPO ligand PK 11195 exhibited a significant protective activity [19,21]. However, the relationship between the affinity of the ligand to TSPO and the pharmacological activity in the various cell lines and the various cellular functions/pathways and models for specific pathological damage is yet unclear. Moreover, the anti-hypoxic effects of PK 11195 are relevant to long-term hypoxia (24 h), but the relevance to a shorter period of hypoxia is yet unclear.

### 4.2. Microglial BV-2 Cells

BV-2 microglial cells were chosen to investigate the effect of CoCl_2_ in a cell line from the central nervous system, which will enable the differentiation of the effects of CoCl_2_ in cell lines from non-neuronal (lung) and central (brain) origin.

CoCl_2_ was used to establish the hypoxia-like condition model in BV2 cells. Similar to the H1299 cells, the impact of CoCl_2_ was concentration-dependent (as assessed at 0.1, 0.3, 0.5, 0.7, 0.9 mM for 30 min, 4 h and 24 h). Such concentration-dependent alterations in cellular processes were also reported in the U118MG glioma cell line [16]. Our findings are consistent with a previous study conducted in our lab on human glioma cells (U118MG cell line), which showed a dose-dependent toxicity of CoCl_2_ [9]. Our data showed that microglial cells derived from the central nervous system may differ from lung cells in their sensitivity to CoCl_2_. After 4 h and 24 h of CoCl_2_ exposure, BV-2 cells exhibited significant reductions in viability in a dose and time-dependent manner. Unfortunately, we did not perform similar experiments in the lung cells. Thus, the dose and time dependency results are relevant only to the glial cells.

In the present study, the CoCl_2_-induced dose-dependent decrease in cellular viability of the BV-2 cell line was demonstrated. Comparable effects were obtained between the alterations which occurred using three concentrations of CoCl_2_ (0.3, 0.5, 0.7 mM) with exposure times of 24 h. The higher concentration of CoCl_2_, namely 0.7 mM, was used in order to obtain more severe hypoxic conditions for the assessment of the protective effects of PK 11195.

In our study using BV-2 cells, we demonstrated an impact of CoCl_2_ exposure on TSPO protein expression levels after 4 h at the chosen concentration. A significant increase in TSPO levels was detected at a concentration of 0.7 mM as compared to the control group (Figure 8). Furthermore, PK 11195 inhibited that reduction in TSPO levels, indicating the possible involvement of TSPO in the hypoxic pathway initiated by in vitro CoCl_2_ exposure.

Similarly, our evaluation of oxidative stress in this cellular model showed a significant elevation after 4 h of exposure to CoCl_2_ at 0.7 mM; however, the pretreatment with PK11195 did not show any protective activity in CoCl_2_-induced oxidative stress. This inconsistency might indicate the lack of involvement of TSPO in pathways leading to oxidative stress (elevation of superoxide and ROS generation) in BV-2 cells. This observation in microglial cell line is in contrast to what was demonstrated in the H1299 cell line, where ROS generation measured by cardiolipin content was prevented by the TSPO ligand PK 11195. The inhibitory effect of PK 11195 on CoCl_2_-induced oxidative stress in H1299 cells was achieved at 0.3 and 0.5 mM CoCl_2_ but not at 0.7 mM, suggesting limited protective capacity of the ligand in preventing excessive oxidative stress.

Additionally, the finding of apoptotic cell death occurrence in H1299 cells and BV2 cells further supports our previous observations [16,21,22]. Apoptosis is a form of programmed cell death, in which caspases are strongly involved. Caspases divided into initiator caspases and executioner caspases. Unfortunately, there was some dissimilarity between experiments in the two cell lines with regard to the super-oxidation and caspase 3. Nevertheless, in the present study, caspase 3 (initiator) and 9 (executioner) were assessed in BV-2 cells as apoptotic markers after exposure to the required duration and concentration of CoCl_2_. The elevation in apoptotic markers was inhibited by PK11195. Interestingly, no significant effect was shown regarding caspase 9 levels. Notably, PK 11195 is a compound with many off targets. Thus, it is possible that some of the findings may not be related directly to TSPO. As previously reported by us, the effects of CoCl_2_ on TSPO expression in the U118MG cell line can be complicated, with a lower concentration actually decreasing, instead of increasing, TSPO protein levels [9]. Thus, concentration–response experiments are required [9]. Moreover, the current study raises a question on the possible role of TSPO in CoCl2 toxicity. It is unclear whether the cellular effects are achieved by direct or indirect ways and whether PK 11195 promotes or inhibits TSPO activity.

In conclusion, CoCl_2_ as a mimicking agent of hypoxia-like conditions, leads to alterations in several apoptosis-associated processes which occur in parallel to a reduction in the levels of TSPO protein levels. Such processes involve essential mitochondrial functions that after a certain time point and at specific CoCl_2_ concentration (0.7 mM), may reach an irreversible damage. The high affinity classical TSPO ligand PK 11195 ligand at a concentration of 25 µM, can prevent some cellular damages caused by exposure to CoCl_2_; however, there are cytotoxic cellular pathways that are insensitive to TSPO ligands and the beneficial effects at present are relevant only to short-term hypoxia-like conditions (24 h). Such pathways, mainly relevant to the generation of oxidative stress, might occur in a non TSPO-related fashion, and thus no inhibitory/protective impact of TSPO ligands can be obtained. Moreover, in future studies, we intend to compare the response of microglial cell lines to macrophage cell lines, both being part of the myeloid system, or to compare the response of glial cells to neuronal cell lines. Nevertheless, hypoxia-like conditions are relevant to both lung cells and brain cells.

## Figures and Tables

**Figure 1 biomolecules-12-01397-f001:**
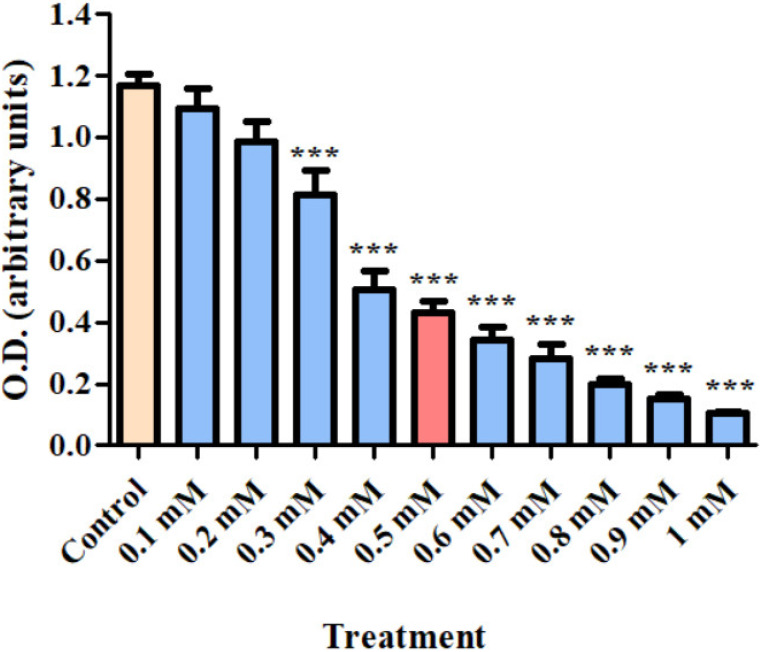
Dose–response analysis of H1299 cell viability (using XTT assay kit) after exposure for 24 h to CoCl_2_ (0.1–1 mM). Results are expressed as mean ± SD (8 replicates for each group). *** *p* < 0.001 compared to control.

**Figure 2 biomolecules-12-01397-f002:**
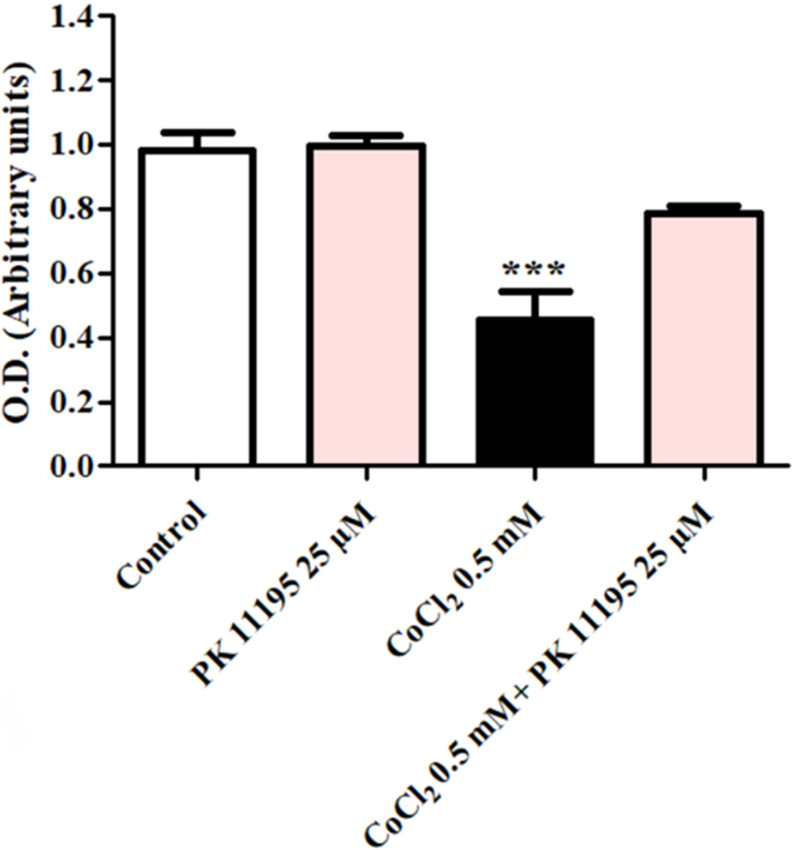
Evaluation of the protective ability of the TSPO ligand PK 11195 to counteract the effect of CoCl_2_-induced cell death. The protective ability of PK 11195 (*n* = 5 replicates) to counteract the effect of 0.5 mM of CoCl_2_ was measured using XTT analysis. Results are expressed as means ± SD ****p* < 0.001 vs. control and vs. CoCl_2_ 0.5 mM+ PK11195 25 µM.

**Figure 3 biomolecules-12-01397-f003:**
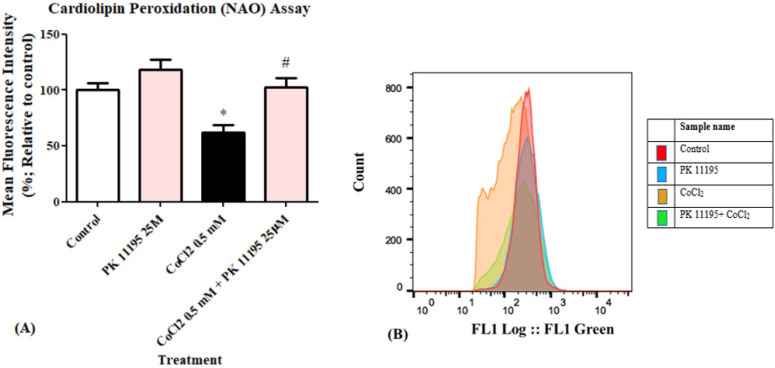
Determination of the efficacy of 25 µM PK 11195 to counteract the effect of 0.5 mM CoCl_2_-induced cardiolipin peroxidation in H1299 cells. The protective efficacy of PK 11195 is shown as (**A**) a bar graph; (**B**) Representative of histogram of the control and CoCl_2_ groups, results are expressed by mean ± SD (*n* = 5 replicates in each group), and (**B**) a histogram. * *p* < 0.05 compared to control and # *p* < 0.05 compared to CoCl_2_.

**Figure 4 biomolecules-12-01397-f004:**
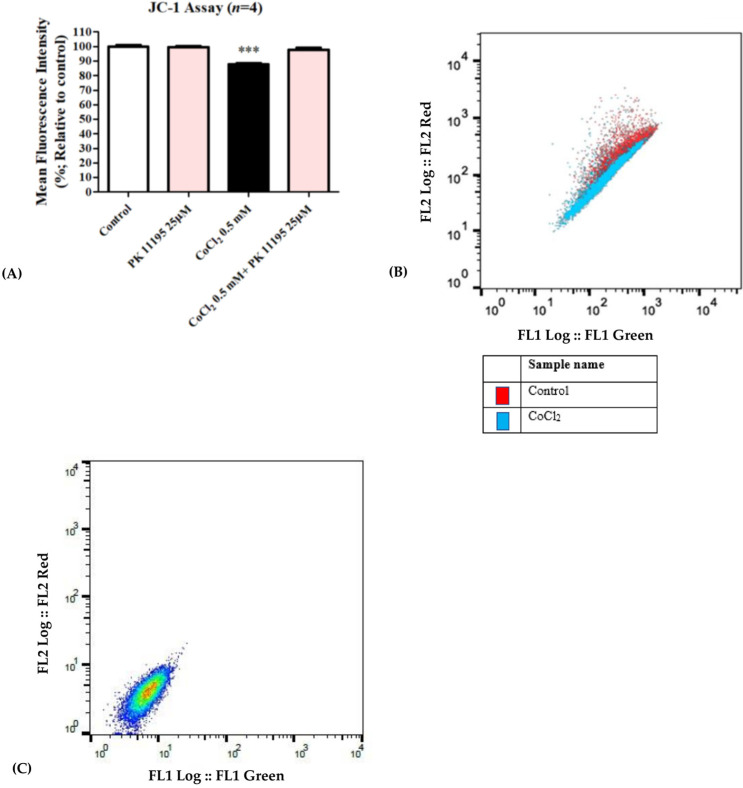
Assessment of mitochondrial membrane potential (JC-1) following exposure of H1299 for 24 h to 0.5 mM CoCl_2_. Inhibition of CoCl_2_-induced mitochondrial membrane potential depolarization by 25 µM PK 11195 expressed as (**A**) a bar graph, and (**B**) Representative of histogram (**C**) Representative of histogram from FlowJo. Results are expressed by means ± SD (*n* = 4 replicates in each group). *** *p* < 0.001.

**Figure 5 biomolecules-12-01397-f005:**
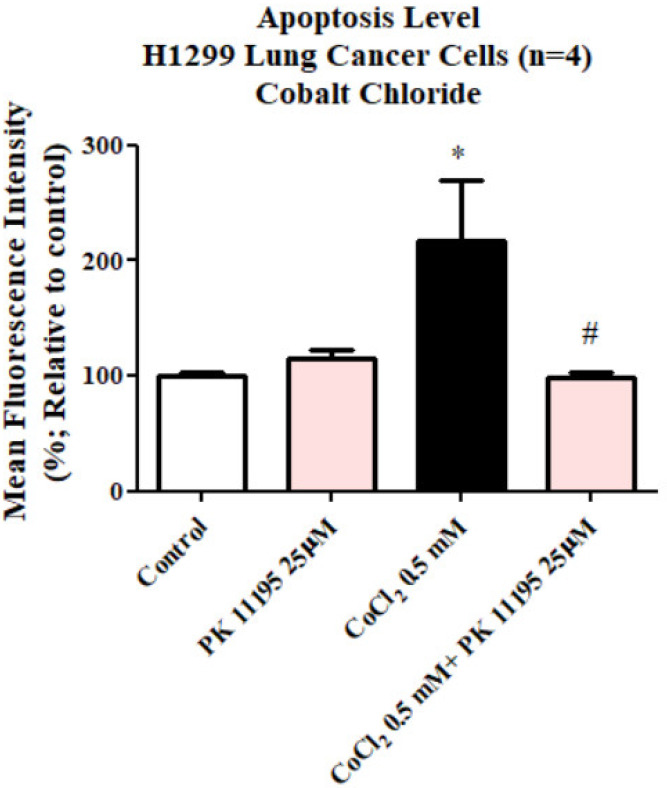
Evaluation of apoptotic cell death levels assessed by apopxin dye following exposure of H1299 for 23 h to 0.5 mM CoCl_2_ with or without pretreatment with 25 µM PK 111195. Results are expressed as mean ± SD (*n* = 4 replicates in each group). # and * *p* < 0.05.

**Figure 6 biomolecules-12-01397-f006:**
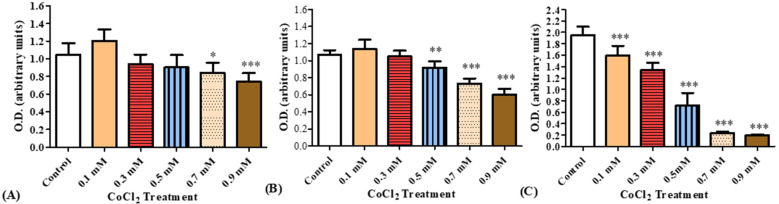
Cellular viability, as measured by XTT, of BV-2 cells exposed to increasing concentrations of CoCl_2_. XTT assay was performed after (**A**) 30 min, (**B**) 4 h, and (**C**) 24 h of CoCl_2_ exposure at concentrations of 0.1, 0.3, 0.5, 0.7, and 0.9 mM. Results are expressed as mean ± SD (*n* = 8 replicates for each group). * *p* < 0.05, ** *p* < 0.01, and *** *p* < 0.001 vs. control.

**Figure 7 biomolecules-12-01397-f007:**
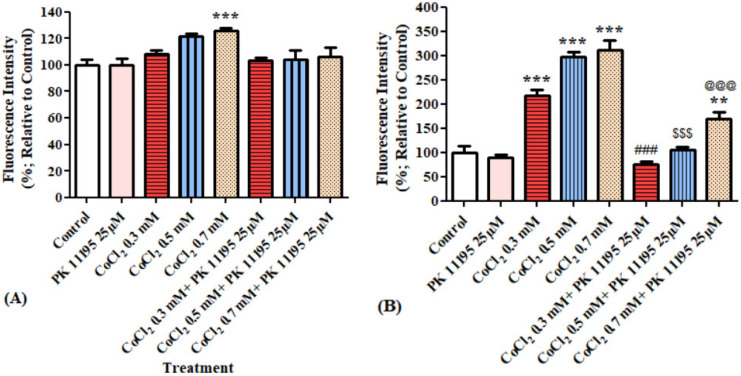
Oxidative stress levels as assessed by ELISA (ROS/superoxide detection assay) following exposure of BV-2 microglial cells to various concentration of CoCl_2_ for 4 h and 24 h with and without pretreatment. Application of PK 11195 (25 µM) as a pretreatment, differentially prevented the elevation of oxidative stress levels caused by the exposure to (**A**) 4 h (*n* = 4 replicates in each group) and (**B**) 24 h (*n* = 5 replicates in each group), CoCl_2_. Results are expressed as mean ± SD. ***p* < 0.01, ****p* < 0.001 vs. control; @@@, $$$, ### vs. corresponding CoCl_2_ concentrations without PK11195.

**Figure 8 biomolecules-12-01397-f008:**
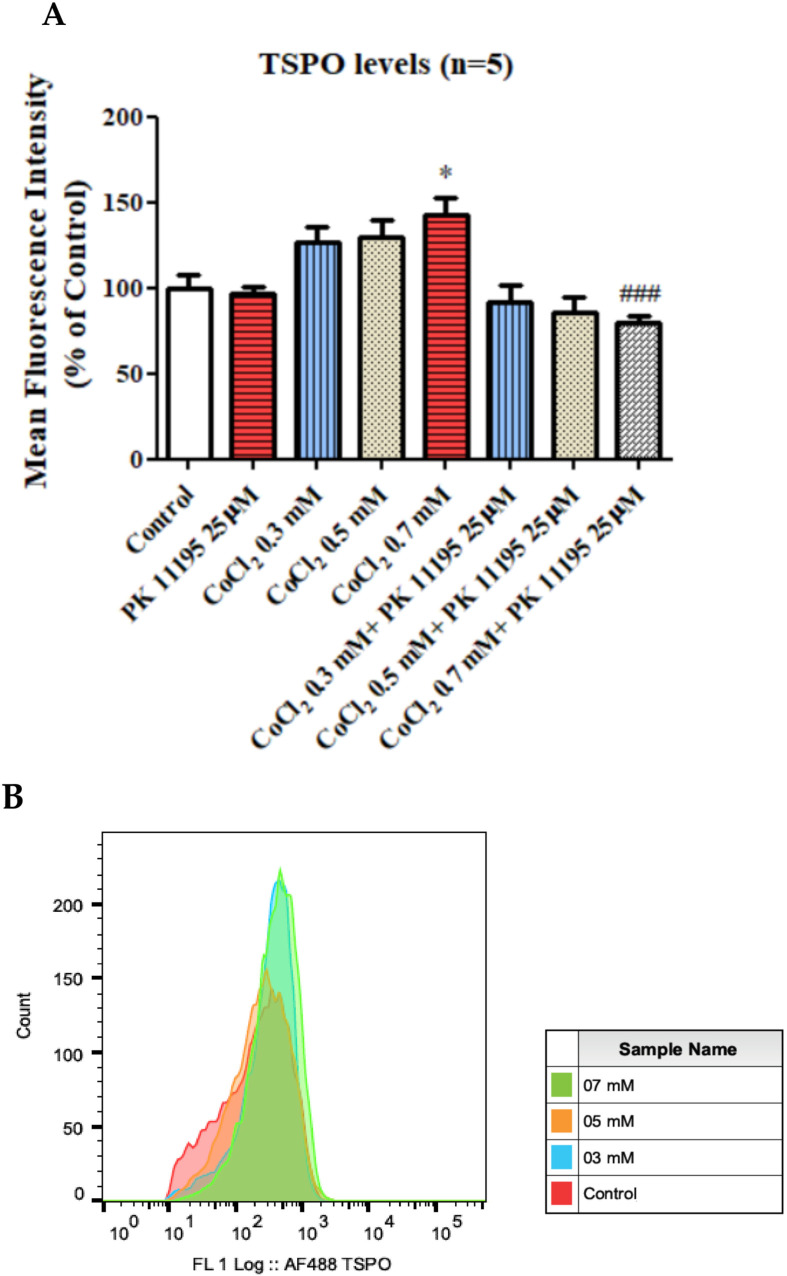
TSPO protein expression levels as assessed by FACS following exposure for 4 h of BV-2 cells to CoCl_2_ at various concentrations with and without PK 11195 pretreatment (25 µM). Alterations in TSPO level following exposure of CoCl_2_ for 4 h (**A**) and representative FACS determination of TSPO levels (**B**). Representative of histogram. Results are expressed as mean ± SD (*n* = 5 replicates for each group). * *p* < 0.05 vs. control and ### *p* < 0.001 vs. CoCl_2_ 0.7 mM.

**Figure 9 biomolecules-12-01397-f009:**
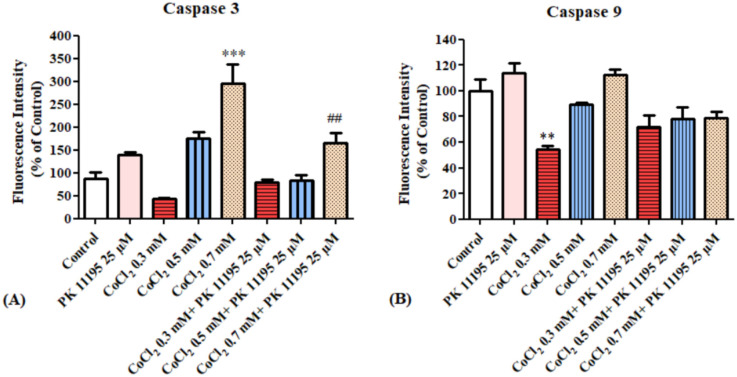
Alterations of caspase 3 and 9 levels following exposure for 4 h of BV-2 microglial cells to various concentrations of CoCl_2_: (**A**) caspase 3 levels, (**B**) caspase 9 levels. Results are expressed as mean ± SD (*n* = 5 replicates for each group). ** *p* < 0.01, *** *p* < 0.001 vs. control, ## vs. CoCl_2_ 0.7 mM.

## Data Availability

Not applicable.

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
