# Peer review of "The Effect of the Classical TSPO Ligand PK 11195 on In Vitro Cobalt Chloride Model of Hypoxia-like Condition in Lung and Brain Cell Lines"

_biomolecules, 2022, doi:10.3390/biom12101397_

Round 1
Reviewer 1 Report
Professor Gavish and his team have produced another article testing TSPO, CoCl2, PK11195. A search in pubmed.org shows 48 citations for "Gavish M and TSPO" only of them include CoCl2 (2009 and 2020), on the other hand 22 citations employ TSPO and PK11195. Professor Gavish's first publication of TSPO was published in 2005, the first publication using PK11195 was in 2006.
My first obstacle to accepting this manuscript is that for a seasoned researcher this article lacks depth. There is no argument for the use of PK11195. PK11195 appeared in the literature in 1983. It was first reviewed in 1984. Since then it remains associated with imaging and very few examples of PK11195 chemical derivatives have been published or resulted in any progress of its use in therapeutics.
In 2015, Su and colleagues (https://pubmed.ncbi.nlm.nih.gov/25514861/) remarked:
[11C]PK11195 has displayed its relatively low brain permeability, low brain plasma protein binding, and poor signal-to-noise ratio, thus, decreasing its sensitivity in detecting microglial activation (Corcia et al. 2012; Venneti et al. 2013). Additionally, its short physical half-life of 20 min and the mutation of the TSPO gene limit the translational utility of [11C]PK11195 as a PET biomarker (Kreisl et al. 2013). Finally, evidence indicated that PK11195 decreases microglial activation, iNOS, IL-1β, IL-6, TNF-a levels and the extent of neuronal damage in quinolinic acid-injected rats (Venneti et al. 2006).
This clearly indicates PK11195 needs modifications to be used in therapies, or to cross the blood-brain barrier. Professor Gavish concludes in this manuscript that:
The novel ligands as well as PK 1195 attenuated the in vitro hypoxia-like effects of CoCl2.It appears that the TSPO ligand PK 11195 can prevent CoCl2-induced cellular damage in both non-neuronal and brain cell lines, and they may offer new therapeutic options in hypoxia-related lung and brain diseases which fail to respond to conventional therapies.
The role of TSPO in apoptosis is known. The role of PK11195 as an inhibitor of TSPO is also known. The role of CoCL2 as mimicking hypoxia is known. The combination of TSPO, PK11195, CoCl2 has been explored by Professor Gavish's laboratory in 2009 and in 2020. This last article was published here in MDPI (Biomedicines).
In 2018 (https://www.ncbi.nlm.nih.gov/pmc/articles/PMC5851033/) researchers describe in the introduction of their study that:
Several specific ligands that modulate TSPO have been described, including benzodiazepines, 1-(2-chlorophenyl)-N-methyl-N-(1-methylpropyl)-3-isoquinolinecarboxamide (PK11195), and N,N-dihexyl-2-(4-fluorophenyl) indole-3-acetamide (FGIN-1-27) (Veenman et al. 2007). Studies have demonstrated that TSPO ligands, including PK11195, may act as agonists or antagonists depending on the ligand concentration and cell type (Totis et al. 1989). Although the underlying mechanisms of action of these TSPO ligands require further clarification, PK11195 and other ligands are currently being used as markers of neuroinflammation in PET imaging (Folkersma et al. 2011), and they also exhibit anticancer (Shoukrun et al. 2008) and immunomodulatory activities (Domingues-Junior et al. 2000). PK11195 has also been shown to increase free radical production in neuronal cells in a TSPO-dependent manner by promoting opening of the mitochondrial permeability transition pore (Jayakumar et al. 2002), and to reduce the proliferation rate of Plasmodium falciparum in infected cells (Dzierszinski et al. 2002, Bouyer et al. 2011).
This article also concludes that :
PK11195 causes the killing of amastigotes in vitro by mechanisms independent of inflammatory mediators and causes morphological alterations within Leishmania parasites, suggestive of autophagy, at doses that are non-toxic to macrophages.
Professor Gavish used 25 micro-molar concentrations (without a reference of the affinity value of TSPo and PK11195) while the researchers above used as much as 400 micro-molar.
Smaller details include, references 3 and 4 are exactly the same. Several of the figures are difficult to visualize (Figure 7A) and in other cases, the format of different panels and the composition are clear misaligned (Figure 8).
The following statement does not correspond to citation 5. Citation 5 does not mention CoCl2
"CoCl2 is a hypoxia-mimicking agent [5]"
Other facts from the literature are missing since they are relevant to the reader (lines 46 and 47): Notably, exposure to CoCl2 is not associated with a decrease in pO2, thus, it actually represents hypoxia-like condition.
This study presents no advancement in the use or modification for use of PK11195 as a therapeutic. The results simply confirm previous studies and do not expand the knowledge accumulated by Professor Gavish's team or the literature. For example, there is not much discussed in the introduction for the reason of choosing the cell lines used in this study or why the U118MG glioblastoma cell line was not used. Thus, the lack of these simple concepts negates the title, which seeks to explain the effects of PK11195 in a hypoxia model of 2 cell lines in vitro but the abstract and discussion conclude that this could translate to therapeutic solutions.
Author Response
Reviewer 1
Professor Gavish and his team have produced another article testing TSPO, CoCl2, PK11195. A search in pubmed.org shows 48 citations for "Gavish M and TSPO" only of them include CoCl2 (2009 and 2020), on the other hand 22 citations employ TSPO and PK11195. Professor Gavish's first publication of TSPO was published in 2005, the first publication using PK11195 was in 2006.
My first obstacle to accepting this manuscript is that for a seasoned researcher this article lacks depth. There is no argument for the use of PK11195. PK11195 appeared in the literature in 1983. It was first reviewed in 1984. Since then it remains associated with imaging and very few examples of PK11195 chemical derivatives have been published or resulted in any progress of its use in therapeutics.
In 2015, Su and colleagues (https://pubmed.ncbi.nlm.nih.gov/25514861/) remarked:
[11C]PK11195 has displayed its relatively low brain permeability, low brain plasma protein binding, and poor signal-to-noise ratio, thus, decreasing its sensitivity in detecting microglial activation (Corcia et al. 2012; Venneti et al. 2013). Additionally, its short physical half-life of 20 min and the mutation of the TSPO gene limit the translational utility of [11C]PK11195 as a PET biomarker (Kreisl et al. 2013). Finally, evidence indicated that PK11195 decreases microglial activation, iNOS, IL-1β, IL-6, TNF-a levels and the extent of neuronal damage in quinolinic acid-injected rats (Venneti et al. 2006).
This clearly indicates PK11195 needs modifications to be used in therapies, or to cross the blood-brain barrier. Professor Gavish concludes in this manuscript that:
The novel ligands as well as PK 1195 attenuated the in vitro hypoxia-like effects of CoCl2.It appears that the TSPO ligand PK 11195 can prevent CoCl2-induced cellular damage in both non-neuronal and brain cell lines, and they may offer new therapeutic options in hypoxia-related lung and brain diseases which fail to respond to conventional therapies.
The role of TSPO in apoptosis is known. The role of PK11195 as an inhibitor of TSPO is also known. The role of CoCL2 as mimicking hypoxia is known. The combination of TSPO, PK11195, CoCl2 has been explored by Professor Gavish's laboratory in 2009 and in 2020. This last article was published here in MDPI (Biomedicines).
In 2018 (https://www.ncbi.nlm.nih.gov/pmc/articles/PMC5851033/) researchers describe in the introduction of their study that:
Several specific ligands that modulate TSPO have been described, including benzodiazepines, 1-(2-chlorophenyl)-N-methyl-N-(1-methylpropyl)-3-isoquinolinecarboxamide (PK11195), and N,N-dihexyl-2-(4-fluorophenyl) indole-3-acetamide (FGIN-1-27) (Veenman et al. 2007). Studies have demonstrated that TSPO ligands, including PK11195, may act as agonists or antagonists depending on the ligand concentration and cell type (Totis et al. 1989). Although the underlying mechanisms of action of these TSPO ligands require further clarification, PK11195 and other ligands are currently being used as markers of neuroinflammation in PET imaging (Folkersma et al. 2011), and they also exhibit anticancer (Shoukrun et al. 2008) and immunomodulatory activities (Domingues-Junior et al. 2000). PK11195 has also been shown to increase free radical production in neuronal cells in a TSPO-dependent manner by promoting opening of the mitochondrial permeability transition pore (Jayakumar et al. 2002), and to reduce the proliferation rate of Plasmodium falciparum in infected cells (Dzierszinski et al. 2002, Bouyer et al. 2011).
This article also concludes that :
PK11195 causes the killing of amastigotes in vitro by mechanisms independent of inflammatory mediators and causes morphological alterations within Leishmania parasites, suggestive of autophagy, at doses that are non-toxic to macrophages.
Professor Gavish used 25 micro-molar concentrations (without a reference of the affinity value of TSPo and PK11195) while the researchers above used as much as 400 micro-molar.
Answer: We have chosen this concentration since PK11195 at this concentration was found to have several cellular activities (line 89-90, reference 10).
Smaller details include, references 3 and 4 are exactly the same. Several of the figures are difficult to visualize (Figure 7A) and in other cases, the format of different panels and the composition are clear misaligned (Figure 8).
Answer: References 3 and 4 were fixed. Figure 7A size was increased for the better visualization. Figure 8 is now aligned.
The following statement does not correspond to citation 5. Citation 5 does not mention CoCl2
"CoCl2 is a hypoxia-mimicking agent [5]"
Answer: Reference was modified.
Other facts from the literature are missing since they are relevant to the reader (lines 46 and 47): Notably, exposure to CoCl2 is not associated with a decrease in pO2, thus, it actually represents hypoxia-like condition.
Answer: An appropriate citation was added (10).
This study presents no advancement in the use or modification for use of PK11195 as a therapeutic. The results simply confirm previous studies and do not expand the knowledge accumulated by Professor Gavish's team or the literature. For example, there is not much discussed in the introduction for the reason of choosing the cell lines used in this study or why the U118MG glioblastoma cell line was not used. Thus, the lack of these simple concepts negates the title, which seeks to explain the effects of PK11195 in a hypoxia model of 2 cell lines in vitro but the abstract and discussion conclude that this could translate to therapeutic solutions.
Answer: Lung and microglial (for brain) cell lines represent cellular response to hypoxia in these cells (line 60-61).
Reviewer 2 Report
The manuscript (Biomolecules-1909336) describes protective effects of the TSPO ligand PK11195 on CoCl2 induced hypoxia-like effects in two cell lines (H1299 lung cell and BV-2 microglia), including cell viability, oxidative stress, mitochondrial damage, apoptosis, and elevated TSPO expression (BV-2 only). It was concluded that TSPO ligands may offer new therapeutic options in hypoxia-related lung and brain diseases. Overall the manuscript is well written. I have the following comments:
1. PK11195 is a very old and dirty compound with many off targets, which might have contributed to much controversy and uncertainty on the exact function of TSPO. I would have expected the authors had employed many other available new generation of ligands more specific to TSPO other than PK11195. Anyway, it should be at least cautioned the many limitations of PK11195 and availability of new TSPO ligands.
2. As previously reported by the authors on U118MG cells, the effects of CoCl2 on TSPO expression can be complicated, with a lower concentration actually decreasing, instead of increasing, TSPO protein levels. This should be acknowledged and discussed in the current manuscript. Further, it does raise the question on possible role of TSPO in CoCl2 toxicity, by direct or indirect ways? And of pk11195, promoting or inhibiting TSPO activity? Can the authors comment on it?
3. Line 395-397: the statement “A significant reduction in TSPO levels was detected at concentration of 0.7 mM as compared to the control group. Furthermore, PK 11195 inhibited that reduction in TSPO levels,…” is opposite to findings in Results (Fig 8).
4. Duplicated ref. 3 and 4;
5. ref 5 (line 40) was not about CoCl2; ref 18, 19 (line 56-57) did not talk about TSPO;
6. There is no part (C) in Fig 3 (line 239-240) – actually a mixed-up of Fig 3 and 4;
7. Ref error in lines 354, 360, 362, 368, 410 – please double check;
8. Fig 7A is corrupted.
Author Response
Reviewer 2
The manuscript (Biomolecules-1909336) describes protective effects of the TSPO ligand PK11195 on CoCl2 induced hypoxia-like effects in two cell lines (H1299 lung cell and BV-2 microglia), including cell viability, oxidative stress, mitochondrial damage, apoptosis, and elevated TSPO expression (BV-2 only). It was concluded that TSPO ligands may offer new therapeutic options in hypoxia-related lung and brain diseases. Overall the manuscript is well written. I have the following comments:
- PK11195 is a very old and dirty compound with many off targets, which might have contributed to much controversy and uncertainty on the exact function of TSPO. I would have expected the authors had employed many other available new generation of ligands more specific to TSPO other than PK11195. Anyway, it should be at least cautioned the many limitations of PK11195 and availability of new TSPO ligands.
Answer: We have performed binding analyses of [3H] PK11195 in H1299 and BV-2 cells (Nagler et al., 2015). We are aware to the problem associated with PK 11195, being a compound with many off targets (line 421-423).
- As previously reported by the authors on U118MG cells, the effects of CoCl2 on TSPO expression can be complicated, with a lower concentration actually decreasing, instead of increasing, TSPO protein levels. This should be acknowledged and discussed in the current manuscript. Further, it does raise the question on possible role of TSPO in CoCl2 toxicity, by direct or indirect ways? And of pk11195, promoting or inhibiting TSPO activity? Can the authors comment on it?
Answer: We thank the reviewer for his comment and this issue is discussed further in lines 423-429.
- Line 395-397: the statement “A significant reduction in TSPO levels was detected at concentration of 0.7 mM as compared to the control group. Furthermore, PK 11195 inhibited that reduction in TSPO levels,…” is opposite to findings in Results (Fig 8).
Answer: We thank the reviewer for his attention and the mistake was addressed and fixed (line 398-400).
- Duplicated ref. 3 and 4;
Answer: The duplication of references was fixed.
- ref 5 (line 40) was not about CoCl2; ref 18, 19 (line 56-57) did not talk about TSPO;
Answer: Yes, the content was changed and it fits references 18-19.
- There is no part (C) in Fig 3 (line 239-240) – actually a mixed-up of Fig 3 and 4;
Answer: Figure 3 legend was modified.
- Ref error in lines 354, 360, 362, 368, 410 – please double check;
Answer: The references were corrected.
- Fig 7A is corrupted.
Answer: Fig 7A was cross verified and it’s now ok.
Round 2
Reviewer 1 Report
Please pay attention to the images.
Figure 3, panel A and B seem to have been prepared differently. The fonts don't match. The same with figure 4. It seems to be a problem with FACS data, you paste the pictures as they come from the FACS software.
Figure 7, panel A does not show (at least in my document). I have attached a document with the way Figure 7 appears on my document.

Author Response
Please pay attention to the images.
Figure 3, panel A and B seem to have been prepared differently. The fonts don't match. The same with figure 4. It seems to be a problem with FACS data, you paste the pictures as they come from the FACS software.
Response: Done
Figure 7, panel A does not show (at least in my document). I have attached a document with the way Figure 7 appears on my document.
Response: Done